# Mapping Genetic Variation in Arabidopsis in Response to Plant Growth-Promoting Bacterium *Azoarcus olearius* DQS-4T

**DOI:** 10.3390/microorganisms11020331

**Published:** 2023-01-28

**Authors:** Fernanda Plucani do Amaral, Juexin Wang, Jacob Williams, Thalita R. Tuleski, Trupti Joshi, Marco A. R. Ferreira, Gary Stacey

**Affiliations:** 1Divisions of Plant Sciences and Technology, C. S. Bond Life Science Center, University of Missouri, Columbia, MO 65211, USA; 2Ginkgo Bioworks, Ag Biologicals, 890 Embarcadero Dr., West Sacramento, CA 95605, USA; 3Department of Electrical Engineering and Computer Science, C. S. Bond Life Science Center, University of Missouri, Columbia, MO 65211, USA; 4Departemnt of BioHealth Informatics, School of Informatics and Computing, Indiana University Purdue University Indianopolis, Indianopolis, IN 46202, USA; 5Department of Statistics, Virginia Tech, Blacksburg, VA 24060, USA; 6Department of Biochemistry and Molecular Biology, Federal University of Parana, Curitiba 19046, Brazil; 7Department of Health Management and Informatics, MU Institute for Data Science and Informatics, C. S. Bond Life Science Center, University of Missouri, Columbia, MO 65211, USA

**Keywords:** plant growth-promoting bacteria_1_ (PGPB), *Arabidopsis thaliana*, natural genetic variation, genome-wide association study (GWAs), agronomic traits

## Abstract

Plant growth-promoting bacteria (PGPB) can enhance plant health by facilitating nutrient uptake, nitrogen fixation, protection from pathogens, stress tolerance and/or boosting plant productivity. The genetic determinants that drive the plant–bacteria association remain understudied. To identify genetic loci highly correlated with traits responsive to PGPB, we performed a genome-wide association study (GWAS) using an *Arabidopsis thaliana* population treated with *Azoarcus olearius* DQS-4^T^. Phenotypically, the 305 Arabidopsis accessions tested responded differently to bacterial treatment by improving, inhibiting, or not affecting root system or shoot traits. GWA mapping analysis identified several predicted loci associated with primary root length or root fresh weight. Two statistical analyses were performed to narrow down potential gene candidates followed by haplotype block analysis, resulting in the identification of 11 loci associated with the responsiveness of Arabidopsis root fresh weight to bacterial inoculation. Our results showed considerable variation in the ability of plants to respond to inoculation by *A. olearius* DQS-4^T^ while revealing considerable complexity regarding statistically associated loci with the growth traits measured. This investigation is a promising starting point for sustainable breeding strategies for future cropping practices that may employ beneficial microbes and/or modifications of the root microbiome.

## 1. Introduction

Plant growth-promoting bacteria (PGPB) benefit plant growth notably by enhancing nutrient uptake, nitrogen fixation, protection from pathogens, stress tolerance and/or boosting plant productivity. Changes in root system architecture and shoot biomass are common plant responses to PGPB, as evidenced on a variety of crops, including maize, rice, wheat, and various bioenergy grasses [1,2,3,4]. Despite an extensive literature documenting the beneficial effect of PGPB on plant growth, we still know relatively little about the molecular details of their mode of action. Genetic variation in the plant host has been reported to modulate the composition of the root-associated microbial population and has been suggested to have important adaptative consequences for plant health [5,6]. The process of domestication has profound consequences on crops, where the domesticate has moderately reduced genetic diversity relative to the wild ancestor across the genome, and severely reduced diversity for genes targeted by domestication [7], which can also impact plant-microbe interaction [8]. Hence, the identification and characterization of genes associated with the capacity of plants to maximize profitable responses to their associated beneficial bacteria could form the basis for breeding approaches to enhance yield, as well as increasing the sustainability of cropping systems. Taking advantage of genetic variation within a natural population, genome-wide association analysis (GWAS) offers the opportunity to analyze associations between single-nucleotide polymorphisms (SNPs) and phenotypic variance, a powerful tool for the identification of genetic loci associated with agronomic traits. For example, GWA mapping analyses have revealed novel and unknown genes impacting a variety of agronomic traits (e.g., flowering time, defense, drought, root cell development, plant architecture, and disease resistance) in numerous plant species, including Arabidopsis [8,9,10], rice [11,12], maize [13,14], and soybean [15,16,17].

Previous studies identified a number of bacterial genes involved in PGPB–plant association, including genes involved in nitrogen fixation [18], siderophore production and iron uptake, phosphate solubilization, production of volatile organic compounds, as well as phytohormones [19,20,21]. There is also a growing realization that plant symbionts can suppress the plant immune system in order to promote their colonization and infection of their plant host [22].

*Azoarcus olearius* DQS-4^T^ is a nitrogen-fixing PGPB with the ability to colonize plant roots and enhance plant growth in monocots, such as rice and *Setaria viridis* [23,24]. In a recent study, we utilized transposon mutagenesis to identify essential bacterial genes that modulated colonization of *Setaria viridis* roots by *A. olearius* DQS-4T and another PGPB, *Herbaspirillum seropedicae* SmR1 [25]. Surprisingly, this study identified very few genes that were critical for both bacteria to colonize *S. viridis* roots. Instead, the data suggest that each bacterium requires a unique set of genes required for root colonization. This genetic diversity in the bacterial partner suggests that a similar level of complexity may exist with regard to how various plant hosts respond to specific PGPB strains. Although only two strains were used in our earlier study, the results appear to be quite distinct from other well-studied, plant–microbial associations. For example, in the rhizobial–legume symbiosis, a core set of both microbial and plant genes appears to be critical for establishment of the symbiosis [26,27,28]. For example, at this point, there appears to be no evidence for the involvement of a common symbiosis pathway (CSP), as defined for the rhizobial and mycorrhizal–plant symbioses, in the intimate association of diverse plant hosts with PGPB [27]. It should be noted that a significant contributor to the identification of the CSP in legumes was the adoption by the research community of model plant species (i.e., *Medicago truncatula* and *Lotus japonicus*) that greatly aided the identification of plant genes essential for establishment of the symbiosis. In a recent review, we argued that the research community interested in PGPB–plant associations would also greatly benefit by adopting model organisms to speed the molecular investigation of these interactions [29].

In the current study, we investigated the phenotypic responses of a natural population of 305 accessions [30] of the model plant *Arabidospsis thaliana* to inoculation by *A. olearius* DQS-4^T^, a PGPB, and performed GWA mapping of four growth traits to identify genetic regions that contribute to bacterial plant growth promotion. The basis of GWAS is the ability to statistically associate specific genetic loci to measured phenotypic diversity within the population and depends on the [31] large linkage disequilibrium (LD) in plants [11,32]. To explore further the output of our analysis, we used two independent, statistical methods to analyze our dataset on the four root growth traits measured. Overlapping SNPs were identified associated with changes in root fresh weight and confirmed by haplotype block analysis. Here, we present the predicted candidate genetic loci from the statistical analysis and discuss their possible relevance to the ability of *A. olearius* DQS-4^T^ to promote plant growth.

GWA mapping resulted in the identification of genetic loci associated with PGPB-induced changes in primary root length or root fresh weight.

## 2. Material and Methods

### 2.1. Bacteria Cultivation

*Azoarcus olearius* DQS-4^T^ [33] was grown overnight at 30 °C on liquid NFbHP-malate modified medium (DL malic acid 20 g L^−1^) supplemented with potassium phosphate (K_2_HPO_4_ 17.8 g L^−1^, KH_2_PO_4_ 159.5 g L^−1^) and ammonium chloride (NH_4_Cl 20 mM) [34,35]. Antibiotics were added to the culture at the following concentrations: 100 µg mL^−1^ streptomycin and 10 µg mL^−1^ nalidixic acid. Subsequently, the DQS-4^T^ culture (OD_600_ = 0.9) was centrifuged at 3000× *g* for 1 min, and the pellet was washed three times by resuspension in 0.9% (*w*/*v*) NaCl. After that, the optical density was adjusted by dilution to 0.005 cells·mL^−1^ (2.3 × 10^5^ CFU·mL^−1^) in 50 mL of NaCl solution.

### 2.2. Plant Growth Conditions and Bacterial Treatment

A collection of 305 natural accessions of *Arabidopsis thaliana* [30] (Appendix A) was used to investigate the response to inoculation by *Azoarcus olearius* DQS-4^T^. Arabidopsis seeds were sterilized by vortexing once with 70% ethanol for 1 min, twice with 70% ethanol plus 0.01% Triton X-100 for 2 min followed by once with 100% ethanol for 2 min. After ethanol removal, seeds were allowed to dry in a sterile hood and then 1 mL sterile water was added prior to vernalization at 4 °C for 3 days. Sterilized seeds were sown on square Petri dishes containing agar-solidified ½ Murashige and Skoog (MS) medium supplemented with 0.5% sucrose and incubated in a vertical position in a plant growth chamber at 21 °C with 16 h light/8 h dark cycle. After 5 days of germination, seedlings of similar size were transferred to circular Petri dishes containing ½ MS agar medium. Plants were inoculated by applying 150 µL of DQS-4^T^ culture containing 2 × 10^5^ cell ml^−1^ onto the agar medium, approximately 5 cm below the root tip, which allowed the roots to grow into the inoculant. The same procedure was done for control treatments using 150 µL of saline solution (0.9% NaCl) without bacteria. The plates were briefly dried in a sterile hood, sealed with parafilm and placed vertically in a growth chamber until phenotypic analysis.

### 2.3. Phenotypic Response of Arabidopsis Accessions to Azoarcus Olearius DQS-4^T^

For each accession, 5 seedlings were grown on a ½ MS agar plate. The reference accessions Col-0 and WS were used in each experiment since they represent non-responding and responding ecotypes, respectively. A total of 3 replicate plates (15 seedlings) of control and DQS-4T-treated seedlings were analyzed for each of the 305 accessions tested. Growth parameters were analyzed 7 days upon treatment by counting lateral root number and measuring primary root length (cm), then average primary root length and lateral root number per seedling were determined. Root and shoot fresh weight were analyzed 8 days after treatment. Data were acquired simultaneously for inoculated and control samples from the three replicates. To determine statistical significance between control and inoculated plants, one-way analysis (ANOVA) was used with Tukey’s test (*p*-value < 0.05). Accessions with significant differences between control and inoculated were categorized as: (A) positive—genotypes that showed trait enhancement due inoculation; (B) negative—genotypes where inoculation inhibited growth; (C) non-responsive—genotypes that showed no growth response to bacterial inoculation.

### 2.4. Data Analysis

In this study, a natural population of 305 accessions of *Arabidopsis thaliana* [30] was used to investigate the genetic basis of the growth response to bacterial inoculation. Mapping analysis was performed on the following root parameters: primary root length (ΔPRL), lateral root number (ΔLRN), root fresh weight (ΔRFW), and shoot fresh weight (ΔSFW). For all traits, means per seedling (n = 5) per biological replicate (n = 3) were used to calculate the mean per treatment per accession. The mean value of the control treatment was subtracted from the value of the inoculated plants to generate the datasets used for GWAS analysis. Genome-wide association analysis employed a mixed linear model (MLM) using Tassel 5.0 software (http://www.maizegenetics.net/tassel, accessed on 15 November 2022) [36] incorporated with population structure by principal component analysis (PCA) and kinship matrix acquired from 1001 Genomes (https://1001genomes.org/, accessed on 15 November 2022) with minor allele frequency (MAF) = as 0.05, and the data were inferred as a normal distribution by the Kolmogorov–Smirnov test. All accessions were genotyped against the Col-0 reference genome with ~214 k single-nucleotide polymorphism (SNPs) markers [37]. Significant SNPs were identified with a strict threshold of significance by Bonferroni correction with *p*-value = 2.34 × 10^−7^. Annotations of candidate genes were retrieved from TAIR10 (http://www.arabidopsis.org, accessed on 15 November 2022). In order to narrow the list of candidate SNPs to a more focused set of SNPs, the data were also analyzed using the two-stages method implemented in the R package GWAS.BAYES [38]. The two steps are called screening and model selection. The screening step of the GWAS.BAYES method performs a usual GWAS analysis with a linear mixed-effects models with a SNP fixed effect and kinship random effects. The screening step provides the usual list of significant SNPs. The model selection step of the GWAS.BAYES method performs a genetic algorithm search through model space, where candidate models are linear mixed-effects models with kinship random effects and may contain multiple SNPs. The genetic algorithm in the model selection step forces the SNPs to compete to appear in the highest ranked models. As shown in [39], combining a screening step and a model selection step provides a much shorter list of significant SNPs and leads to a much higher true discovery rate.

Manhattan plots and linkage disequilibrium (LD) plots were generated using the R statistical software 4.0 [40].

### 2.5. Validation Using Quantitative Reverse Transcription PCR (qRT-PCR)

Gene expression was evaluated by qRT-PCR. RNA was isolated from roots 7 days after mock or bacterial treatment using Direct-zol RNA kit treated with DNase (Zymo Research, Irvine, CA, USA) following the manufacturer’s instructions. cDNA synthesis was performed with M-MLV reverse transcriptase (Promega, Madison, WI, USA). qRT-PCR was carried out as follows: 10 ng cDNA, 3 pM of each primer and PowerUp^TM^ SYBR^TM^ Green master mix (Applied Biosystems, Waltham, MA, USA) were mixed and amplified in Applied Biosystems ABI PRISM^TM^ 7500 detection system (Applied Biosystems). Three biological replicates and three technical replicates for each transcript were analyzed using LinReg PCR 11.1 [41]. Quantitative amplifications were performed for different genes and ubiquitin 10 (At4G05320) was used as an internal reference. Primers used are listed in Appendix A.

## 3. Results

### 3.1. Arabidopsis Thaliana Response to Azoarcus olearius DQS-4^T^ Inoculation

Previous, published research, including from our own laboratory [23], documented that *A. olearius* DQS-4^T^ can produce strong, positive effects on root growth in both rice and *Setaria viridis* [24]. As a prelude to our larger GWAS study, we initially tested only a few Arabidopsis ecotypes regarding their response to DQS-4^T^ inoculation. These initial experiments revealed that *A. thaliana* ecotype Columbia (Col-0) had no measurable response to bacterial inoculation, while ecotype Wassilewskija (Ws) showed a robust and significant increase in all the traits measured (lateral root number, root and shoot fresh weight) (Appendix A). Although limited, these initial experiments were important in showing phenotypic diversity in the plant response to inoculation, as well as providing both a negative (Col-O) and positive (Ws) control for future experiments.

### 3.2. Natural Variation in the Response of Arabidopsis Accessions to A. olearius Inoculation

In order to investigate the natural variation in the responsiveness of Arabidopsis to *A. olearius* DQS-4^T^, a total of 305 Arabidopsis accessions were analyzed for changes in root and shoot traits upon bacterial treatment. Analysis of correlation to measure the direction and strength between control and DQS-4^T^-treated samples showed a low correlation coefficient between control and treated samples for shoot fresh weight, primary root length and root fresh weight (R^2^ = 0.452, R^2^ = 0.349 and R^2^ = 0.106, respectively). However, for lateral root number, the correlation was only slightly stronger (R^2^ = 0.501) between control and treated samples (Figure 1A–H), suggesting that the magnitude of these DQS-4^T^-induced growth responses were weakly related to the intrinsic growth capacity for these parameters under the tested conditions. Hence, faster-growing accessions or accessions that form more lateral roots in the experimental setup are not necessarily stronger responders to bacterial treatment.

### 3.3. Response Categories of Arabidopsis thaliana Growth Traits to Treatments

The measured, phenotypic variability, relative to lateral root number (LRN), primary root length (PRL), root and shoot fresh weight (RFW, SFW), among the 305 Arabidopsis accessions classified into 3 categories are: 1. positive responsive: accessions that demonstrate a significant, positive change upon inoculation compared to mock samples; 2. non-responsive: accessions where bacterial treatment did not affect growth; 3. negative responsive: accessions where treatment inhibited growth (Figure 2). Within the positive responsive category, eleven genotypes showed growth enhancement in all four parameters analyzed, where most of the changes were statistically significant in lateral root number followed by shoot fresh weight and root fresh weight (Figure 2A,C). The growth of 13 ecotypes clearly benefited from DQS-4^T^ inoculation (Figure 2B,C). For instance, the genotype Ws showed an increase in biomass and root growth when inoculated (Figure 2C).

However, the largest number of ecotypes (129) fell within the non-responsive group, which showed no statistically significant positive or negative response to bacterial inoculation for the parameters analyzed (Figure 2D–F). As mentioned above, Col-0 is a good example of an ecotype within this non-responsive category (Figure 2F). Interestingly, 82 genotypes showed a negative growth response to bacterial inoculation (Figure 2G). An example is ecotype PHW-35 (Figure 2H–I), which showed a negative response in each of the four parameters measured. However, when present, the positive effects on root architecture can be dramatic (Figure 3); for example, as shown in Figure 3A for ecotype Kr-0 (Figure 3A).

Other ecotypes, such as Ws, CIBC-4, Si-0, Hodvala-2 and wag-3, demonstrated similar increases in root fresh weight upon bacterial treatment (Figure 3B) and could be easily identified from non-responsive ecotypes, such as Col-0, DUK, N7, Sap-0, Zdrl2-24 and MIB-15 (Figure 3C,D). However, it is important to note that it was common to find ecotypes that were not consistently responsive for all parameters measured (Figure 3E,F). For instance, ecotype Aa-0 showed a significant increase in LRN upon inoculation but was non-responsive for the other root and shoot parameters. Another example of trait-related variation is exemplified by ecotype Hod which showed no significant changes to LRN and PRL; however, RFW was significantly reduced by inoculation while increasing shoot biomass. This variability within overall response, responses in individual parameters, and opposing responses (i.e., negative, and positive) suggests significant underlying complexity in the genetics of the plant response, as well as the molecular mechanisms involved. However, unlike Aa-0 and Hod, some ecotypes gave robust responses for all four traits analyzed, including ecotypes Ws, Bla-1, KI-5, Kr-0 or showed a consistent, negative response, such as ecotypes UKNW-06-460, UKSE 06-349, PHW-35 and PHW-37 (Table 1 and Appendix A for a complete dataset).

### 3.4. Genome-Wide Association Loci Mapping in the Arabidopsis Population

The dataset collected for DQS-4^T^-induced changes in lateral root number (∆LRN), primary root length (∆PRL), root fresh weight (∆RFW) and shoot fresh weight (∆SFW) were averaged and analyzed against the Col-0 reference genome using a mixed linear model (MLM) algorithm in Tassel 5.0 Software. To determine the distribution and quality of the data within the population a quantile-quantile plot (Q-Q plot) was generated for each growth trait (Appendix A). The GWA mapping results showed highly significant SNPs for two traits ∆PRL and ∆RFW (Figure 4). No significant SNPs were significantly associated to ∆LRN and ∆SFW (Appendix A). With a threshold of −log_10_(*P*) > 7 adjusted by Bonferroni correction, a total of 63 loci were detected for root fresh weight and 55 loci correlated to primary root length (Appendix A). We observed only one SNP associated with both traits ∆RFW and ∆PRL, mapping close to the gene encoding AtFKGP, bifunctional fucokinase/fucose pyrophosphorylase (At1G01220, *p*-value = 2.19 × 10^−8^ and 3.32 × 10^−7^, adjusted by Bonferroni correction, respectively). Given this close association, we measured the expression of *AtFKGP* by qRT-PCR using mRNA extracted from roots of 7-day-old seedlings of ecotypes representing non-responsive (Col-0, N7), responsive (Ws, Kr-0 and CIBC-4), and negative responding lines (PHW-37 and Lis-1) either mock non-inoculated or inoculated with DQS-4^T^. However, this experiment failed to find either down or upregulated gene expression of *AtFKGP* in response to bacterial inoculation in any of the Arabidopsis accessions tested (Appendix A).

### 3.5. Primary Root Length Highly Correlated SNPs and Candidate Genes

We identified 55 SNPs highly correlated to measured changes in PRL (Appendix A). An example are SNPs mapping close to gene At1G33410, encoding the suppressor of auxin resistance 1 (*sar1*) gene. SAR1 and SAR3 are proteins similar to vertebrate nucleoporins that are part of the nuclear pore complex (NPC). Plants deficient in either protein exhibit pleiotropic growth defects partially affecting the translocation of proteins involved in hormonal signaling and plant development [42,43]. A large LD resulted in inclusion of many polymorphisms in this candidate region; for example, loci within a gene predicted to encode a tetratricopeptide repeat 9 (TPR9) protein involved in gibberellic acid regulation [44], fascinated stem 4 (Atfas4) protein and a Ring/Ubox super family (At1g01660) protein. Moreover, a SNP highly correlated to primary root length corresponded to the gene encoding a late elongate hypocotyl LHY1, a MYB-related putative transcription factor implicated in circadian regulation of flowering time [45].

### 3.6. Root Fresh Weight Highly Correlated SNPs and Candidate Genes

We identified 65 SNPs highly correlated to changes in RFW Appendix A). In order to narrow the list of candidate SNPs to a more focused set of SNPs, the data were also analyzed using the two-stages procedure implemented in the R package GWAS.BAYES [38,39]. This analysis reduced the number of SNPs highly associated with ∆RFW to 11 (Table 2); however, no SNPs were highly associated with ∆PRL in this analysis. In most GWA studies, the highly trait-correlated SNPs can present alleles in linkage disequilibrium (LD) at two or more loci in a population. However, because the peak of selection signals is relatively large in the GWA peak region, it is difficult to conclude whether the target of selection is the causative SNP or other alleles are significantly associated with this genome region. To examine the relationship between SNPs and regions, a haplotype block was generated for 10 kb upstream or downstream of the most significantly associated SNP at position 13,459,922 on chromosome 4 observed for ∆RFW (Figure 5A,B). The selected SNP 13,459,922 (*p* = 7.55 × 10^−9^; At4G26690) mapped close to a glycerophosphodiester phosphodiesterase-like 3 GDPDL3 gene, also known as Shaven 3 (SHV3) that is involved in cell wall organization and root hair growth [46]. This SNP was identified as significant in both statistical methods used. SNP 13,459,922 is located at an intronic 5′ untranslated region (5′UTR) with a modifier predicted effect [47]. Next, we determined which gene in proximity to this highly associated genomic region underlies the variation. As shown in Figure 5, six haplotype blocks were significantly associated with SNP 13459922, At4G26690. Based on haplotype analysis, seven SNP regions were identified associated, respectively, with genes encoding a member of the vacuolar-type ATPase family (At4G26710) and a protein phosphatase x-1(PP4) (At4G26720), extensin-like protein (Lip5, At4G26750), microtubule-associated protein 65-2 (MAP65-2, At4G26760), phosphatidate cytidylyltransferase (CDS3, At4G26770), mitochondrial GRPE 2 (At4G26780) and Lipase acyl hydrolase superfamily (GDSL-like, At4G26790) (Figure 5C). Because the polymorphisms are difficult to identify, we also carried out a matrix analysis, that showed SNPs at position 13,477,249 (gene Lip5) and 13,483,356 (CDS3 gene) highly correlated to SNP 13459922, GDPDL3 corroborating the haplotype block (Appendix A). Next, we determined the expression level of the genes correlated to the GWA peak by quantitative RT-PCR (qRT-PCR) using mRNA isolated from root tissue. We assumed that the expression of a candidate gene might be altered in accessions of different responsive categories. Hence, we extracted mRNA from the roots of selected accessions responsive (Ws and Kr-0), non-responsive (Col-0 and N7) and negative response (PHW-37 and Lis-1) regarding the RFW trait. The data show that none of the accessions tested showed a significant change in expression level upon inoculation for the various genes tested (i.e., those encoding GDPDL3, Lip5 and CDS3) (Appendix A). Of course, the lack of a transcriptional response to bacterial inoculation does not rule out the possibility that a specific gene could be playing an important role in the response to *A. olearius* DQS-4^T^ treatment.

## 4. Discussion

The benefits of PGPB in promoting plant growth, improving nutrient uptake and plant resilience to biotic and abiotic stress, and boosting crop production are documented by an expansive literature [3,48,49,50]. While the molecular mechanisms and specific pathways that underlie the growth-promoting responses in plants by PGPB have been investigated to some extent regarding the bacterial functions, left largely unexplored are the plant functions involved. Better defining these functions is important since they may help address the problems of consistency and efficiency that are found commonly when PGPB are used under field conditions to enhance crop yield and sustainability [51,52,53]. GWAS is now a popular method to harness natural genetic variation in a population to identify genetic loci critical for specific agronomic traits and in support of breeding improvement programs [13,54,55,56]. Although used with great success in many studies, classical GWAS relying on SNPs has its limitations due to ‘missing heritability’ [57]. Failure to capture rare variants, allelic heterogeneity, epistasis, and/or epigenetic variation often decreases the detecting capacity of GWAS [58,59,60,61,62,63]. To test the feasibility of this approach to investigate PGPG–plant interactions, we applied GWAS to map loci within the model plant Arabidopsis crucial for the beneficial response to the PGPB *Azoarcus olearius* DQS-4^T^. Such an approach has been used previously; for example, to examine the response of an *Arabidopsis* natural population to inoculation with the PGPB *Pseudomonas fluorescens.* This study identified 10 potential genes candidates involved in changes root architecture and shoot biomass but found none strongly correlated to growth responses to bacterial inoculation with no common gene that could be correlated to a PGPB mediated effect [21].

A few general conclusions can be made from our study. Consistent with published reports, including some from our own studies [2,23,64] plant genotype largely determines whether a given PGPB strain will or will not enhance or inhibit plant growth. This could be a major factor in field-to-field variation in published PGPB studies [51,52,53]. However, perhaps most impactful, is that the data point to considerable complexity in the mechanisms that underlie a beneficial plant response to PGPB inoculation. For example, within the Arabidopsis population, 27% of ecotypes showed no response to inoculation, while others showed either a negative (12% PRL, 4% LRN, 6% RFW and 6%SFW) or positive (8% PRL, 13% LRN, 10% RFW and 11% SFW) response to a specific trait. Considering the four growth parameters tested, 11 ecotypes showed consistently positive response whereas 6 accessions responded negatively. Indeed, some ecotypes showed different responses regarding a specific phenotypic parameter. For instance, ecotype Hod showed increased shoot fresh weight while root biomass was negatively affected. This complexity correlates well with our recent, mutational analysis of two PGPB strains that suggested that the gene functions necessary for plant root colonization are unique to a given strain, with only a few genes appearing essential for both strains tested. This large variation, coupled with normal issues found when applying biological inoculation to cropping systems, could, in large part, explain why it is not uncommon to find very variable, inconsistent results when PGPB are used under field conditions [51,52,53].

The power of the GWAS method is the ability to statistically correlate specific genetic loci with the phenotypic variation measured across the entire population [65]. Hence, we used two, orthogonal methods for data analysis with increased statistical stringency. For example, our initial analysis using a mixed-effects model identified several candidate SNPs associated with changes in primary root length and root fresh weight. However, in root length none of those SNPs were significant when submitted to a stricter statistical analysis. We encountered two major challenges to identify statistically significant SNP associations with the phenotypes measured. First, although several statistically robust models have been developed, false positives can still arise from population structure. Second is the large extended, linkage disequilibrium (LD) which results in the inclusion of many candidate genes within a single LD block, making it if very hard to ultimately identify the underlying, causative gene. For example, this might explain why none of the candidate genes associated with ∆PRL were found to respond transcriptionally to PGPB inoculation.

Among the candidate genes identified were a phosphate transporter essential for arbuscular mycorrhizal symbiosis [66] and glutamine synthetase (GS) known for its role in assimilation of nitrogen and biosynthesis of glutamine in plants. Transcriptional levels of the GS2 gene showed no changes in sugarcane leaves inoculated with several PGPB bacteria strains. However, these authors did detect biochemical differences relative to concentrations of glutamine and glutamate [67]. The ankyrin repeat family protein, was previously found in a study that characterized genes that contribute to bacterial adaptation isolated from Arabidopsis, maize, and poplar trees [68]. In rice, ankyrin-repeat protein was shown to be upregulated in response to rice leaf bright pathogen *Xanthomonas oryzae* (*xoo*) suggesting its role as a positive regulator in basal defense pathways [69].

Unfortunately, ecotype Col-0 did not respond to inoculation for any of the growth parameters measured. Hence, all the genetic resources available for this ecotype were of little use for follow-up genetic studies. For example, the gene encoding Lyst-interacting protein 5, (LIP5) was within the haplotype associated with ∆RFW. Arabidopsis LIP5 is part of the endosomal sorting complexes required for transporters (ESCORTs) for sustained protein trafficking. Lyst-interacting protein 5 (LIP5) interacts with MAP kinase 3 (MPK3) and MPK6 in response to pathogen infection, playing a critical role in plant basal resistance [70,71]. Mutational disruption of LIP5 expression had little effect on pathogen associated molecular pattern (i.e., flagellin) or salicylic acid-induced defense responses but compromised basal resistance in response to the bacterial pathogen *Pseudomonas syringae* [70]. We obtained the available T-DNA insertion line of AtLIP5 within the Col-O background and found, not surprisingly, that this mutation had no effect on the response of seedlings to *A. olearius* DQS-4^T^ inoculation (data not shown). Hence, confirmation of those candidate genes implicated by our GWAS analysis in the PGPB response awaits the ability to use gene-editing to create mutations in those specific ecotype backgrounds that do respond to inoculation.

In summary, our study of the natural genetic variation within an Arabidopsis population showed considerable variation in the ability of plants to respond to inoculation by *A. olearius* DQS-4^T^ while revealing considerable complexity regarding statistically associated loci with the growth traits measured and in the patterns (positive, neutral, negative) of those responses. Considering all candidate genes with SNP–trait associations in the GWA analysis, several have known or predicted functions that hold promise for being functional in mediating PGPB growth effects.

## Figures and Tables

**Figure 1 microorganisms-11-00331-f001:**
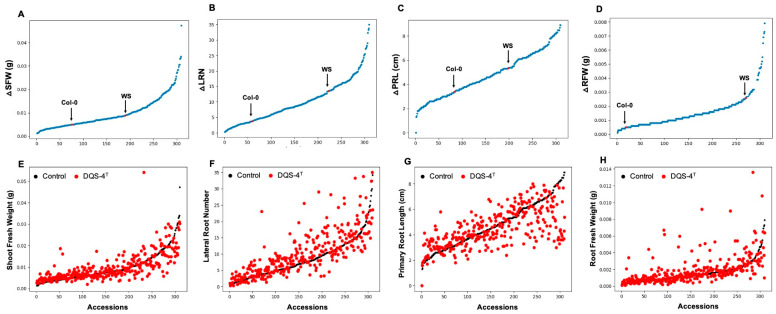
Natural variation in 305 *A. thaliana* accessions in response to the plant growth-promoting *Azoarcus olearius* DQS-4^T^. (**A**) Accessions sorted for increase in shoot fresh weight (∆SFW) in response to DQS-4T (Col-0 and Ws are indicated with black or red arrow dot). (**B**) Accessions sorted for increase in lateral root number (∆LRN) in response to DQS-4T. (**C**) Accessions sorted for increase in primary root length (∆PRL) in response to DQS-4T. (**D**) Accessions sorted for increase in root fresh weight (∆RFW) in response to DQS-4T. (**E**) Average shoot fresh weight (∆SFW) of control (black dots) and DQS-4T (red dots) plants. (**F**) Number of lateral roots (∆LRN) formed in control (black dots) and DQS-4T-treated (red dots) plants. (**G**) Primary root length (∆PRL) of control (black dots) and DQS-4T (red dots) plants. (**H**) Average root fresh weight (∆RFW) of control (black dots) and DQS-4T (red dots) plants. Each dot represents the average of three biological replicates.

**Figure 2 microorganisms-11-00331-f002:**
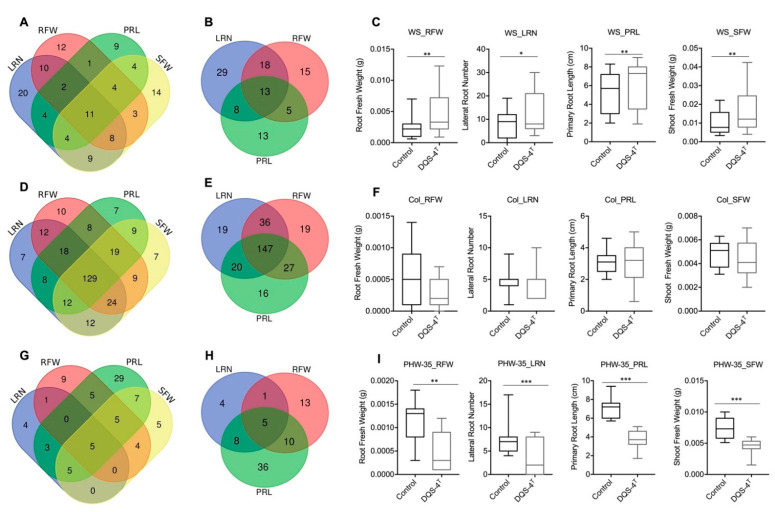
Growth responses of control and inoculated Arabidopsis accessions. Venn diagram showing the total number of common accessions across traits in each response category. (**A**) Responsive accessions to DQS-4T inoculation for root and shoot traits. (**B**) Responsive accessions in root growth parameters. (**C**) Significant response of Wassilewskija (WS) genotype upon inoculation with DQS-4T in all four traits. (**D**) Venn diagram of non-responsive accessions to DQS-4T inoculation for each root and shoot traits. (**E**) Significant non-responsive accessions in root growth parameters. (**F**) No significant response of Col-0 upon inoculation with DQS-4T in all four traits. (**G**) Accessions that responded negatively to DQS-4T inoculation in each trait analyzed. (**H**) Negative response accessions in root growth parameters. (**I**) Significant response of PHW-35 control upon inoculation with DQS-4T in all four traits analyzed. Bars are an average of three biological replicates (n = 15). Statistical analysis was carried out using one-way ANOVA with Tukey’s test *p*-value = 0.05 (* *p* ≤ 0.05; ** *p* ≤ 0.01; *** *p* ≤ 0.001).

**Figure 3 microorganisms-11-00331-f003:**
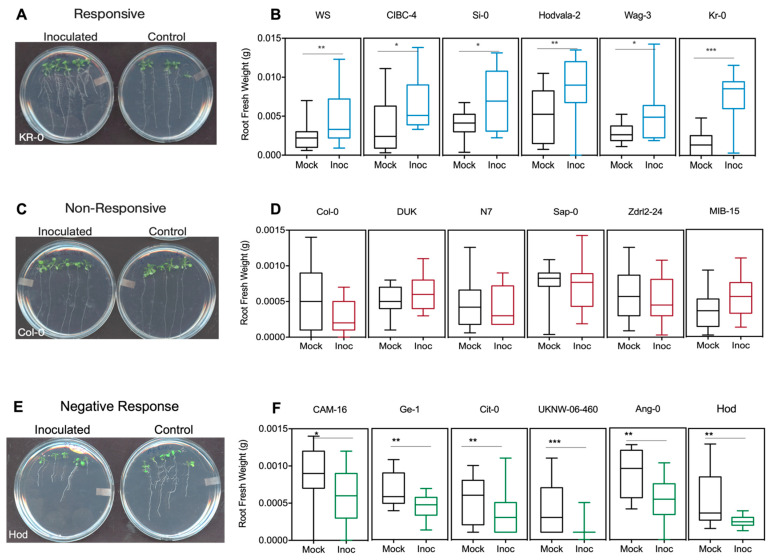
Phenotyping of Arabidopsis accessions for responsiveness to DQS-4T mediated effect in root fresh weight (RFW). Agar plates with five seedlings showing the different response of Arabidopsis accessions to *Azoarcus olearius*. (**A**,**B**) Accessions that increased RFW upon DQS-4T treatment. (**C**,**D**) Non-responsive accessions to bacteria treatment. (**E**,**F**) Accessions where RFW was inhibited by DQS-4T treatment. Bars are an average of three biological replicates (n = 15). Statistical analysis was carried out using one-way ANOVA with Tukey’s test *p*-value < 0.05 (* *p* ≤ 0.05; ** *p* ≤ 0.01; *** *p* ≤ 0.001). RFW = root fresh weight. Photographs were taken by scanning the plates using a photo scanner with resolution 640 × 480.

**Figure 4 microorganisms-11-00331-f004:**
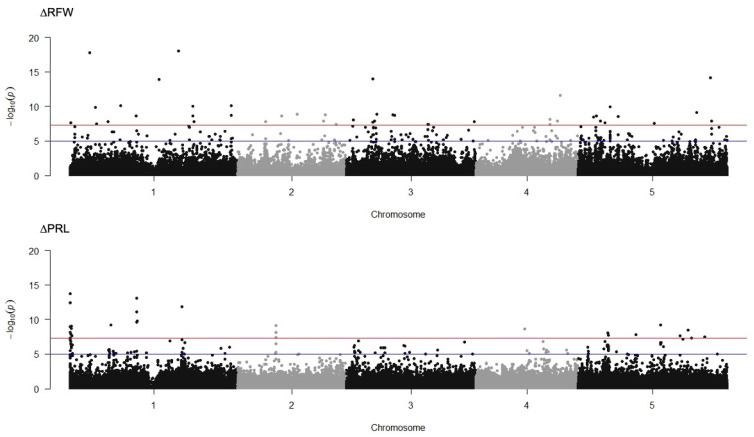
Manhattan plot of GWA mapping of *A. olearius* DQS-4^T^ effects on root fresh weight (∆RFW) and primary root length (∆PRL) in −log_10_(*P*). SNP marker trait associations are shown as black and grey dots for each chromosome. The red and blue lines indicate the arbitrary thresholds of −log10(*P*) = 5 and −log10(*P*) = 7.

**Figure 5 microorganisms-11-00331-f005:**
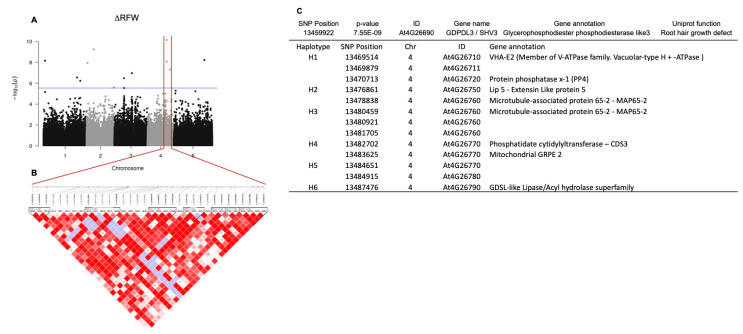
(**A**). Manhattan plot (−log_10_(*P*)) of a genome-wide association study (GWAS) of 305 Arabidopsis accessions treated with *Azoarcus olearius* DQS-4T. The GWAS significance level was set at 2.34 × 10^−7^ and plotted as a red line. (**B**). Haplotype physical location around the highly correlated SNP At4G26690. (**C**). Haplotype analysis of alleles in linkage disequilibrium (LD) with a highly correlated SNP of root fresh weight (∆RFW).

**Table 1 microorganisms-11-00331-t001:** Genotypes with statistical significance for the four plant growth parameters measured with regard to DQS-4^T^ treatment.

		LRN	RFW	PRL	SFW
Response	Genotype	*p*-Value			
Positive	WS	2.22 × 10^−3^	9.60 × 10^−3^	1.34 × 10^−2^	4.67 × 10^−3^
	Bla-1	3.27 × 10^−5^	3.56 × 10^−4^	3.40 × 10^−4^	4.81 × 10^−6^
	KI-5	8.21 × 10^−3^	5.13 × 10^−2^	1.83 × 10^−2^	3.94 × 10^−2^
	Ka-0	7.67 × 10^−4^	2.30 × 10^−2^	3.51 × 10^−4^	9.94 × 10^−3^
	LDV-25	5.90 × 10^−13^	4.02 × 10^−8^	8.31 × 10^−11^	2.17 × 10^−8^
	HS-0	3.92 × 10^−3^	1.59 × 10^−3^	6.93 × 10^−3^	1.72 × 10^−3^
	DralV-15	2.65 × 10^−3^	9.26 × 10^−5^	1.12 × 10^−17^	1.52 × 10^−5^
	In-0	3.00 × 10^−3^	3.63 × 10^−2^	2.21 × 10^−3^	1.34 × 10^−2^
	Hodvala-2	2.22 × 10^−2^	5.20 × 10^−3^	7.74 × 10^−3^	1.66 × 10^−3^
	Kr-0	2.36 × 10^−4^	5.51 × 10^−6^	6.12 × 10^−4^	1.10 × 10^−4^
	JEA	9.02 × 10^−4^	1.87 × 10^−2^	3.63 × 10^−3^	9.04 × 10^−5^
Negative	PHW-35	2.87 × 10^−3^	9.19 × 10^−5^	5.88 × 10^−10^	1.01 × 10^−5^
	PHW-37	4.90 × 10^−5^	1.55 × 10^−3^	8.45 × 10^−9^	8.30 × 10^−6^
	UKSE 06-349	1.65 × 10^−4^	1.56 × 10^−4^	1.76 × 10^−4^	4.12 × 10^−4^
	UKNW-06-460	7.31 × 10^−5^	8.33 × 10^−3^	3.26 × 10^−5^	9.82 × 10^−6^
	Lis-1	3.33 × 10^−2^	3.59 × 10^−2^	3.12 × 10^−5^	1.50 × 10^−4^

**Table 2 microorganisms-11-00331-t002:** List of candidate genes from GWAs analysis of the *A. olearius* DQS-4^T^ -mediated plant effects on root fresh weight (∆RFW) of 305 *Arabidopsis thaliana* population.

*Chr*	Candidate Gene	Loci Position	*p*-Value	Gene Annotation
1	At1G03530	882791	8.32 × 10^−8^	Nuclear assembly factor 1 (ATNAF1)
1	At1G10660	3534853	1.75 × 10^−18^	Transmembrane protein
1	At1G14040	4812798	3.23 × 10^−8^	PHO1 homolog 3
1	At1G22550	7967378	4.68 × 10^−7^	NPF5.16
1	At1G52710	19638846	9.13 × 10^−19^	Rubredoxin-like superfamily protein
2	At2G18245	7939481	2.18 × 10^−9^	alpha/beta-Hydrolases superfamily protein
3	At3G14400	4812265	1.84 × 10^−8^	Ubiquitin-specific protease 25
4	At4G14820	8507871	1.15 × 10^−7^	Pentatricopeptide repeat (PPR) superfamily protein
4	At4G26690	13459922	7.55 × 10^−9^	Glycerophosphodiester phosphodiesterase-like 3 (GDPDL3)
5	At5G08640	2804242	3.19 × 10^−9^	Flavonol synthase 1
5	At5G35630	13833427	2.63 × 10^−8^	Glutamine synthetase 2
5	At5G60070	24191284	1.57 × 10^−7^	Ankyrin repeat family protein

## Data Availability

Not applicable.

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
