# Peer review of "Mapping Genetic Variation in Arabidopsis in Response to Plant Growth-Promoting Bacterium Azoarcus olearius DQS-4T"

_microorganisms, 2023, doi:10.3390/microorganisms11020331_

Round 1

Reviewer 1 Report

The authors described a genome wide association mapping analysis for genetic determinants that drive the plant-bacteria association. In general, the article is written in a clear and understandable way and the structure and figures are easy to follow. The experiments are repeated sufficiently. I only have a few minor suggestions:

1. The raw data for the root phenotypes of 305 accessions should be provided as a supplemental file.

2. Figure 2C,F,I, I would suggest presenting data in a consistent format.

3. Figure 3, scale bars for pictures are missing.

4. For the method parts, minor allele frequency (MAF) or minor allele count (MAC) should be provided; The tests for data normality are required.

Author Response

Dear reviewer, we appreciate your comments and made the changes accordingly.  

1. The raw data for the root phenotypes of 305 accessions should be provided as a supplemental file.

Answer: The raw data for the root phenotype is provided as Supplementary Material S1 at the results section in Response categories of Arabidopsis thaliana growth traits to treatments.

2. Figure 2C,F,I, I would suggest presenting data in a consistent format.

Answer: The graphs for LRN were modified to be consistent with the other parameters and replaced.

3. Figure 3, scale bars for pictures are missing.

Answer: Instead of scale bars the resolution of the pictures was added to the figure legend with a brief description. “Photographs were taken by scanning the plates using a photo scanner with resolution 640 x 480.

4. For the method parts, minor allele frequency (MAF) or minor allele count (MAC) should be provided; The tests for data normality are required.

Answer: We have added the MAF to the material and methods lines 197-198 “…minor allele frequency (MAF) is setting as 0.05, and the data is inferred as a normal distribution by Kolmogorov-Smirnov test.”

Reviewer 2 Report

Dear authors, the manuscript entitled "Mapping genetic variation of Arabidopsis in response to Plant Growth Promoting Bacterium Azoarcus olearius DQS-4T" presents an interesting research approach on the topic plant-microorganism interaction. The concept of plant growth promotion is important for the improvement of future agronomic techniques and the increase of sustainability.

There are some suggestions that I consider will improve your work.

The Abstract and Introduction are well written and clear. The information provided in Introduction section sustains the research conducted by the authors.

The last paragraph of the Introduction should be reorganized. I suggest you to present only the Aim and the objectives/hypotheses in this paragraph. Make them clear, in one sentence each. All the information related to other research should be moved to the Discussion section or within the text of the Introduction. 

Material and Methods section - Add a sub-section with Data analysis, where you can describe the methods and why did you choose them for analysis and the package used. This is important for other researchers to replicate your study.

 The Results and Discussion are well organized and written. Data resulted from the research is presented in a clear manner, and the connections with other studies are pointed well.

Overall, the article is interesting and provides a good basis for future studies in the field of plant growth promotion. 

Author Response

Dear reviewer, we acknowledge your suggestions and modified accordingly.

1. The last paragraph of the Introduction should be reorganized. I suggest you to present only the Aim and the objectives/hypotheses in this paragraph. Make them clear, in one sentence each. All the information related to other research should be moved to the Discussion section or within the text of the Introduction.

Answer: The paragraph was modified as suggested:  “In the current study, we investigated the phenotypic responses of a natural population of 305 accessions [30] of the model plant Arabidospsis thaliana to inoculation by A. olearius DQS-4T, a PGPB, and performed GWA mapping of four growth traits to identify genetic regions that contribute to bacterial plant growth promotion. The basis of GWAS is the ability to statistically associate specific genetic loci to measured phenotypic diversity within the population and depends on the [31] large linkage disequilibrium (LD) in plants [11, 32]. To explore further the output of our analysis we used two independent, statistical methods to analyze our dataset on the four root growth traits measured. Overlapping SNPs were identified associated with changes in root fresh weight and confirmed by haplotype block analysis. Here, we present the predicted candidate genetic loci from the statistical analysis and discuss their possible relevance to the ability of A. olearius DQS-4T to promote plant growth”.

2. Material and Methods section - Add a sub-section with Data analysis, where you can describe the methods and why did you choose them for analysis and the package used. This is important for other researchers to replicate your study.

Answer: We acknowledged your suggestion and modified the section to Data analysis. We have included a clear explanation of the methods used to select significant SNPs. At first the analysis was performed by TASSEL version 4.0. TASSEL is a widely accepted software for GWAS studies using Mixed Linear Model algorithm. The first analysis generated an extensive list of candidate SNPs. To find a more specific list of SNPs we used GWAS.BAYES method. We have included the most recent publication about GWAS.BAYES published by our collaborators in 2022 (J. Williams, M.A.R. Ferreira and T. Ji (2022), BICOSS: Bayesian iterative conditional stochastic search for GWAS, BMC Bioinformatics 23, 475. https://doi.org/10.1186/s12859-022-05030-0).

203 – 214 “In order to narrow the list of candidate SNPs to a more focused set of SNPs, the data were also analyzed using the two-stages method implemented in the R package GWAS.BAYES [38]. The two steps are called screening and model selection. The screening step of the GWAS.BAYES method performs a usual GWAS analysis with a linear mixed effects models with a SNP fixed effect and kinship random effects. The screening step provides the usual list of significant SNPs. The model selection step of the GWAS.BAYES method performs a genetic algorithm search through model space, where candidate models are linear mixed effects models with kinship random effects and may contain multiple SNPs. The genetic algorithm in the model selection step forces the SNPs to compete to appear in the highest ranked models. As shown in [39], combining a screening step and a model selection step provides a much shorter list of significant SNPs and leads to a much higher true discovery rate.

Reviewer 3 Report

The manuscript reports on the mapping of genetic variation of Arabidopsis in response to Plant 2 Growth Promoting Bacterium Azoarcus olearius DQS-4T

the comments below aim to be constructive

line 397 'For example, within the Arabidopsis population, most ecotypes showed no response to inoculation, while others showed either a negative of positive response.' - a somewhat vague statement, can the percentage of each response type be cited here

line 399 'Only a small, subset of eco-types showed consistently positive or negative responses for the four growth parameters tested and, indeed, some ecotypes showed a positive or neutral response regarding a spe-cific phenotypic parameter, while showing a negative response in other parameters.' - can the authors expand on this statement and explain any similarities between the ecotypes in this subset that might determine the consistency in response?

Author Response

Dear reviewer, we acknowledge your suggestions and changed the manuscript accordingly.

A sentence with the percentage of each response was added to the discussion covering both lines mentined above.

472-479 “For example, within the Arabidopsis population, 27% of ecotypes showed no response to inoculation, while others showed either a negative (12% PRL, 4% LRN, 6% RFW and 6%SFW) or positive (8% PRL, 13% LRN, 10% RFW and 11% SFW) response to a specific trait. Considering the four growth parameters tested, 11 ecotypes showed consistently positive response whereas 6 accessions responded negatively. Indeed, some ecotypes showed different responses regarding a specific phenotypic parameter. For instance, ecotype Hod showed increased shoot fresh weight while root biomass was negatively affected.”